# Effects of Heat Stress on Gut-Microbial Metabolites, Gastrointestinal Peptides, Glycolipid Metabolism, and Performance of Broilers

**DOI:** 10.3390/ani11051286

**Published:** 2021-04-30

**Authors:** Guangju Wang, Xiumei Li, Ying Zhou, Jinghai Feng, Minhong Zhang

**Affiliations:** State Key Laboratory of Animal Nutrition, Institute of Animal Sciences, Chinese Academy of Agricultural Sciences, Beijing 100193, China; 82101185163@caas.cn (G.W.); llxiumei93@163.com (X.L.); 15624955881@163.com (Y.Z.); fjh6289@126.com (J.F.)

**Keywords:** heat stress, microbial metabolites, gastrointestinal peptides, glycolipid metabolism

## Abstract

**Simple Summary:**

In the summer, heat stress is a main factor that causes poor performance in broilers. Broilers are more susceptible to high temperature environments than mammals due to their lack of sweat glands and being covered in feathers. Heat stress can alter the regulation of glycolipid metabolism, which is manifested by unstable levels of blood glucose, insulin, total cholesterol, and triglyceride. Heat stress also affects the structure of gut microbes and gastrointestinal peptides. However, the relationship among microbiota, gastrointestinal peptides, glycolipid metabolism, and production performance under heat stress is still unclear. Moreover, exploring these mechanisms can help in the development of strategies that alleviate the negative effects of performance by heat stress. Our results suggest that the poor production performance of broilers under heat stress may be related to short chain fatty acids fermented by intestinal microbiota involved in regulating metabolic disorders.

**Abstract:**

This paper investigated the effects of heat stress on gut-microbial metabolites, gastrointestinal peptides, glycolipid metabolism, and performance of broilers. Thus, 132 male Arbor Acres broilers, 28-days-old, were randomly distributed to undergo two treatments: thermoneutral control (TC, 21 °C) and high temperature (HT, 31 °C). The results showed that the average daily gain (ADG), average daily feed intake (ADFI), and gastric inhibitory polypeptide (GIP) concentration in the jejunum significantly decreased the core temperature, feed conversion ratio (FCR), and ghrelin of the hypothalamus, and cholecystokinin (CCK) in jejunum, and serum significantly increased in the HT group (*p* < 0.05). Exploration of the structure of cecal microbes was accomplished by sequencing 16S rRNA genes. The sequencing results showed that the proportion of *Christensenellaceae* and *Lachnospiraceae* decreased significantly whereas the proportion of *Peptococcaceae* increased at the family level (*p* < 0.05). *Ruminococcus* and *Clostridium* abundances significantly increased at the genus level. Furthermore, the content of acetate in the HT group significantly increased. Biochemical parameters showed that the blood glucose concentration of the HT group significantly decreased, and the TG (serum triglycerides), TC (total cholesterol), insulin concentration, and the insulin resistance index significantly increased. Nonesterified fatty acid (NEFA) in the HT group decreased significantly. In conclusion, the results of this paper suggest that the poor production performance of broilers under heat stress may be related to short-chain fatty acids (SCFAs) fermented by intestinal microbiota involved in regulating metabolic disorders.

## 1. Introduction

Chicken meat is considered an important source of dietary protein worldwide. As per the Food and Agriculture Organization (FAO), total global chicken production was 118.0 million tons in 2019, accounting for a large proportion of the whole meat production. Broilers are severely affected by heat stress; it can cause slow growth rate, low feed intake, and decreased immunity, leading to economic losses [1,2]. Studies have shown that heat stress leads to glucolipid metabolism disorder [3,4,5]. Heat stress accelerates decomposition of glycogen and inhibits glycogen production [6]. When ambient temperature is higher than the tolerable limit of broilers, the energy produced cannot meet the needs, resulting in insulin resistance and glucolipid metabolism disorders [7]. Previous studies investigated the impacts of heat stress on broilers, which was found to cause blood glucose instability [3,8,9], decrease insulin sensitivity [10], serum cholesterol, and triglycerides [11], increase body fat in the abdomen, and decrease plasma non-esterified fatty acid (NEFA) concentrations [12]. Recently, several studies have shown that heat stress affects the metabolites and structures of the intestinal microflora [13,14] and gastrointestinal peptide [15] in broilers. However, to date, there are no specific studies on the effects of heat stress on broiler microflora and its metabolites, gastrointestinal peptide, glycolipid metabolism, and the association among the three.

Multiple studies were conducted on the associations among bacterial metabolites, gastrointestinal peptides, and glucose and lipid metabolism on metabolic disorders in mice and humans. Intestinal microbiota are involved in metabolic processes and energy homeostasis [16]. Previous studies have shown that heat stress can alter the gut microbiota of mice [17]. The Firmicutes to Bacteroidetes ratio decreased significantly under 30 °C [18]. Short-chain fatty acids (SCFAs) are metabolites formed by gut microbes from complex dietary carbohydrates. Researchers found that SCFAs stimulated mouse L cells to produce glucagon-like peptide 1 (GLP-1), peptide YY (PYY), and other intestinal anorexia hormones, reducing appetite [19]. GLP-1 and GIP are related to insulin secretion, and approximately 70% of β-cell insulin secretion is controlled by GIP and GLP-1 [20]. Insulin secretion disorders lead to metabolic disorders, such as diabetes and obesity [21]. As mentioned above, heat stress causes metabolic disorders in broilers. However, the association between metabolic disorder and microbiota under heat stress is still unclear in mice and humans.

Thus, the purpose of this study was to investigate the effects of heat stress on the microbiota and its metabolites, gastrointestinal peptides, and glycolipid metabolism in broilers, and explore the relationship among them to provide a scientific basis for reconstructing the intestinal flora, to alleviate the decline in production performance caused by heat stress.

## 2. Materials and Methods

### 2.1. Animals and Experimental Design

A total 132 male, one day old, Arbor Acres broilers were purchased from commercial hatcheries and housed in three-layer (8400 cm^2^ per layer) metal cages at ambient temperatures that decreased with age. All birds had free access to feed and water (room temperature) ad libitum. The broilers were fed on crumble diets (Table 1). Then, birds were divided into a thermoneutral control group (TC, 21 °C) or a high temperature group (HT, 31 °C), with six biological replicates per group, 11 birds per replicate, at 21 days old. Birds were transferred to the environmental temperature control chamber, while maintaining a temperature of 21 °C, a humidity of 60%, for 7 days. The experiment started at 28 days old. Temperature control was adjusted by an artificial environmental control chamber developed by the Institute of Animal Science of the Chinese Academy of Agricultural Sciences (CAAS). There was no significant difference in the initial body weight of broilers in the two groups. The temperature of the two groups was constant and humidity remained at 60% until the end of the experiment, which lasted for 21 days. The lighting program was continuous throughout the experimental (fluorescent light, 40 W). To reduce stress, irrelevant personnel were prohibited from entering the artificial environment control chamber. This program was approved by the Experiment Animal Welfare and Ethical at the Institute of Animal Science of CAAS.

### 2.2. Sample and Data Collection

During the experiment, we used an 0.01 g sensitive electronic body weight scale (manufacturer: Mettler Toledo, PL2002) to record the initial feed weight, final feed weight, and body weight of the bird replicates, then calculated the average feed intake, average body weight, average daily weight gain, and feed efficiency for each replicate. This experiment used a rectal probe thermometer to measure the body core temperature of one bird replicate, which was randomly selected from each replicate, and measured four times a day during the experiment. At the end of the experiment, one bird, after 12 h fasting, was randomly selected from each replicate for insulin and blood glucose determination. Feed then continued for two hours, and one bird was randomly selected from each replicate to collect samples. The blood was immediately collected through the wing vein into heparinized tubes and centrifuged at 10,000× *g* for 4 min at 4 °C. Then the plasma was collected and stored at −20 °C until the analysis of the concentrations of PYY, ghrelin, CCK, GIP, and GLP-1. Immediately after the blood samples were obtained, the birds were humanely sacrificed, and the tissues of the cecum contents, intestinal mucosa, pancreas, and hypothalamus were collected. The samples were placed in a cryopreservation tube and stored in a −80 °C refrigerator.

### 2.3. Determination of Gastrointestinal Peptides

The concentrations of CCK, Ghrelin, GLP-1, GIP, and PYY in the intestinal mucosa and serum were determined by the enzyme-linked immunosorbent assay (ELISA). The intestinal mucosa needed to be grinded, weighed, diluted with PBS buffer at 1:9, and centrifuged for 20 min (2000–3000 rpm). The supernatant was then carefully taken for testing. The standard wells were set up, the samples added (diluent and enzyme label reagent in sequence), and were incubated for 60 min. The ELISA plate was washed with a washing solution for more than five times, and the color reagent and stop solution were added. Finally, an enzyme-labeled instrument was used to measure the absorbance of each well (OD value). The contents of each peptide in the sample were calculated through the standard curve.

### 2.4. DNA Extraction and PCR Amplification

Microbial community genomic DNA was extracted from cecal content samples using the E.Z.N.A.^®^ soil DNA Kit (Omega BioTek, Norcross, GA, USA) according to the manufacturer’s instructions. The DNA extract was checked on 1% agarose gel, and DNA concentration and purity were determined with NanoDrop 2000 UV-vis spectrophotometer (Thermo Scientific, Wilmington, DE, USA). The hypervariable region V3-V4 of the bacterial 16S rRNA gene was amplified with primer pairs 338F (5′-ACTCCTACGGGAGG-CAGCAG-3′) and 806R (5′-GGACTACHVGGGTWTCTAAT -3′) by an ABI GeneAmp^®^ 9700 PCR thermocycler (Applied Biosystems, Carlsbad, CA, USA). The PCR amplification of 16S rRNA gene was performed as follows: initial denaturation at 95 °C for 3 min, followed by 27 cycles of denaturing at 95 °C for 30 s, annealing at 55 °C for 30 s and extension at 72 °C for 45 s, and single extension at 72 °C for 10 min, and end at 10 °C. The PCR mixtures contain 5 × TransStart FastPfu buffer 4 μL, 2.5 mM dNTPs 2 μL, forward primer (5 μM) 0.8 μL, reverse primer (5 μM) 0.8 μL, TransStart FastPfu DNA Polymerase 0.4 μL, template DNA 10 ng, and finally ddH2O up to 20 μL. PCR reactions were performed in triplicate. The PCR product was extracted from 2% agarose gel and purified using the AxyPrep DNA Gel Extraction Kit (Axygen Biosciences, Union City, CA, USA), according to manufacturer’s instructions, and quantified using Quantus™ Fluorometer (Promega, Madison, WI, USA).

### 2.5. Illumina MiSeq Sequencing

Purified amplicons were pooled in equimolar and paired-end sequences on an Illumina MiSeq PE300 platform/NovaSeq PE250 platform (Illumina, San Diego, CA, USA), according to the standard protocols by Majorbio Bio-Pharm Technology Co. Ltd. (Shanghai, China).

### 2.6. Processing of Sequencing Data

The raw 16S rRNA gene sequencing reads were demultiplexed, quality-filtered by fastp version 0.20.0 [22] and merged by FLASH version 1.2.7 [23] with the following criteria: (i) the 300 bp reads were truncated at any site receiving an average quality score of <20 over a 50 bp sliding window, and the truncated reads shorter than 50 bp were discarded. Reads containing ambiguous characters were also discarded. (ii) Only overlapping sequences longer than 10 bp were assembled according to their overlapped sequences. The maximum mismatch ratio of overlap region was 0.2. Reads that could not be assembled were discarded. (iii) Samples were distinguished according to the barcode and primers, and the sequence direction was adjusted, exact barcode matched, and two nucleotides mismatched in the primer matching. Operational taxonomic units (OTUs) with 97% similarity cutoff [24,25] were clustered using UPARSE version 7.1 [24], and chimeric sequences were identified and removed. The taxonomy of each OTU representative sequence was analyzed by RDP Classifier version 2.2 [26] against the 16S rRNA database using confidence threshold of 0.7.

### 2.7. Determination of SCFAs

We accurately weighed 1 g of cecal content, added 5 mL of ultra-pure water, shook, and mixed for 30 min, overnight at 4 °C, then centrifuged at 10,000 rpm for 10 min, and transferred the supernatant to a 10 mL cuvette. We added 4 mL of ultra-pure water to the precipitation, shook and mixed for 30 min, then centrifuged at 10,000 rpm to transfer the supernatant into a 10 mL cuvette for constant volume. We transferred the liquid in the colorimetric tube to a 10 mL centrifuge tube, centrifuged at 12,000 rpm for 15 min, and then transferred the supernatant to a 2 mL centrifuge tube, according to V:V = 9:1 (900 μL supernatant + 100 μL 25% metaphosphoric acid), mixed well, and let it stand at room temperature for 3–4 h centrifugation, 45 um microporous membrane (nylon series) filtration. We added the machine bottle (more than 600 μL) to be tested. Chromatographic conditions: db-ffap column, 30 m * 250 μm * 0.25 μm; carrier gas: high purity nitrogen (99.999%), flow rate: 0.8 mL/min; auxiliary gas: high purity hydrogen (99.999%), detector FID temperature: 280 °C, injection port temperature: 250 °C, split ratio: 50:1, injection volume: 1 μL; temperature programming: initial temperature: 60 °C, the temperature rose to 220 °C at the rate of 20 °C/min, and held for 1 min.

### 2.8. Determination of Serum Biochemical Parameters

The concentrations of blood glucose, serum insulin, triglycerides, total cholesterol, and NEFA were measured using the kits provided by Nanjing Jiancheng Bioengineering Institute (Jiangsu, China). The insulin resistance index was calculated using a formula as previously described: insulin resistance index = insulin/(22.5e^-lnglucose^) [27].

### 2.9. Statistical Analysis

All statistical analyses were conducted using GraphPad Prism 8 software (GraphPad Software, Inc. La Jolla, CA, USA). An independent sample t-test (unpaired Student’s *t* test and Mann–Whitney test) was used for the comparison of the two treatments. Replicate served as the experimental unit. The confidence interval was 95%, and *p* < 0.05 indicated a significant difference in the treatment effect; values were expressed as the mean ± SEM.

## 3. Results

### 3.1. Effect of High Temperature on Performance

The effect of heat stress on performance is shown in Table 2. Compared with the TC group, ADG and ADFI in the HT group decreased significantly (*p* < 0.05). Compared with the TC group, the core temperature of birds and FCR in the HT group significantly increased (*p* < 0.05).

### 3.2. Effects of High Temperature on Gastrointestinal Peptide

The effects of heat stress on the neuropeptides of gut and hypothalamus concentrations in six-week-old chickens were examined and the results are presented in Figure 1, Figure 2 and Figure 3. Compared with the TC group, GIP concentration in the jejunum of the HT group significantly decreased (*p* < 0.05). The concentration of CCK in the jejunum and serum and ghrelin of the hypothalamus in the HT group was significantly higher than that in the TC group (*p* < 0.05). The GLP-1 in the ileum, PYY in the pancreas, and ghrelin in the jejunum in the HT group were not significantly different from that of the TC group.

### 3.3. Effects of High Temperature on Cecal Microbial Composition

The intestinal microbial community of broilers is mainly concentrated in the cecum; it is the main place where microorganisms participate in the regulation of body metabolism. Therefore, we analyzed the composition of the caecal flora of two groups of broilers at the levels of phylum, family, and genus. The compositions of cecal microbiota at phylum, family, and genus levels are provided in Table 3. At the phylum level, the two groups of flora were mainly composed of *Firmicutes*, *Bacteroides*, and *Actinomycetes*; however, there was no significant difference in abundance. At the family level, the dominant bacteria were mainly *Ruminococcaceae*; the proportion of *Christensenellaceae* and *Lachnospiraceae* decreased significantly, and increased the proportion of *Peptostreptococcaceae* (*p* < 0.05). It may show that heat stress significantly affects the dominant intestinal flora at the family level. At the genus level, the dominant bacteria were mainly *Faecalibacterium*, *Romboutsia*. *Ruminococcus* and *Clostridium* abundances significantly increased, indicating that the thermal environment significantly increased the proportion of them.

### 3.4. Effects of High Temperature on SCFAs Concentration

The concentration of short chain fatty acids in cecal contents of the two treatment groups is shown in Table 4. Compared with the TC group, the content of acetate in the HT group significantly increased, but there was no significant difference in propionic acids and butyric acids.

### 3.5. Effects of High Temperatures on Serum Parameters

The blood glucose, insulin, insulin resistance index, TG, TC, and NEFA concentrations are shown in Figure 4. Compared with the TC group, the blood glucose and NEFA concentrations in the HT group significantly reduced, the insulin, TC, and TG concentrations significantly increased. Compared with the TC group, the insulin resistance index of broilers under heat stress increased significantly, resulting in severe insulin resistance.

## 4. Discussion

Thermal environment severely impairs the performance [28,29], changes the composition of gut microbiota [30], and affects glycolipid metabolism [31] of broilers. However, previous studies have not been able to account for the associations among them. The present study clearly established a significant outcome of a thermal environment on performance, gut microbiota, gastrointestinal peptides, and glycolipid metabolism in broilers, and developed a preliminary understanding of the relationship among them.

Significant elevations were detected in the core temperatures of the birds, meaning that heat stress was produced. Previous literature demonstrated that heat stress could negatively impact ADFI, ADG, and FCR in broilers [2,29,32,33,34]. The present study showed that ADFI, ADG, and FCR were seriously affected by high temperatures of 31 °C compared with the 21 °C group, which agreed with the results of previous studies [2,35]. These results are widely accepted, they imply that heat exposure will directly impair productive performance of birds. In general, the reason for the decline in productive performance is largely due to the lower feed intake; birds reduce feed intake to minimize excess metabolic energy production to keep themselves cool [36]. Another important reason for the weight gain reduction observed in birds experiencing heat stress would be metabolism disorder. Previous studies have shown that thermal environment can disrupt metabolic homeostasis, accelerating protein catabolism [37], increasing abdominal fat deposition [38], changing blood glucose levels, decreasing insulin sensitivity, and causing a negative energy balance [39]. Birds suffering from heat stress may choose to use more energy to resist the damage caused by high temperatures rather than to grow or reproduce. Therefore, the underlying cause of poultry production performance degradation may be metabolic disorder. The current study observed that heat stress led to elevated levels of insulin, serum triglycerides, and total cholesterol, and a reduction in serum blood glucose and NEFA. Changes in blood glucose in broilers exposed to high temperatures showed mixed results, as there was evidence for it to increase [3], decrease, or remain unchanged [9]. Numerous factors influence blood glucose and the reasons for the differences include different physiological statuses and experimental designs. The results of elevated levels of insulin [40], serum triglycerides, total cholesterol, and reduced NEFA [41] in this study are consistent with previously reported studies. It is possible that insulin levels were elevated due to compensatory secretions to maintain glucose tolerance. The increased basal insulin levels may explain the lack of an increase in basal NEFA levels because insulin is a potent antilipolytic hormone. We also calculated that the insulin resistance index increased significantly, meaning insulin resistance occurs under heat stress. In summary, these observations imply that heat stress alters carbohydrate and lipid metabolism, leading to metabolic disorder.

Gut microbes have previously been susceptible to heat stress in both mice and livestock. The present study found that heat stress changed the colony composition at the family level, reduced the proportions of *Christensenellaceae* and *Lachnospiraceae*, and increased the proportion of *Peptococcaceae*. It was reported that *Christensenellaceae* is related to healthy glucose metabolism and reduces the risk of obesity [42]. A previous study reported gut dysbiosis in mice that underwent chronic water avoidance stress; in particular, the abundance of *Lachnospiraceae* declined [43]. Similar results were obtained in this paper. The increase of *Peptococcaceae* was mainly related to histopathological infection, indicating that heat stress increased the threat factors of gut health. At the genus level, heat stress increased the proportion of *Ruminococcus* and *Clostridium*. In addition, our results showed that heat stress increased the content of acetate in cecal contents. Since *Ruminococcus* and *Clostridium* are the main acetate-producing microorganisms, the increase of acetate content in cecal contents may be associated to the increase of *Ruminococcus* and *Clostridium*.

Gastrointestinal peptides are mainly secreted by intestinal endocrine cells, including ghrelin, CCK, GIP, PYY, GLP-1, which are directly involved in the regulation of gastrointestinal peristalsis, sensation, and secretion. They have the dual functions of promoting the secretion of neurotransmitters and hormones, and regulating feeding behavior, nutrient absorption, energy metabolism, intestinal peristalsis, and emptying [44]. The results we obtained show that heat stress significantly increased CCK concentration in the jejunum and circulation—similar to the results obtained by a previous study [45]. CCK proved to be an anorexic hormone in various poultry studies, which can gather short-term post-eating satiety signals and inhibit broiler feeding [46]. It was reported that exogenous injection of CCK could significantly reduce the feed intake of poultry [47]. Therefore, the significant decrease in feed intake of broilers in the HT group may be due to the large amount of CCK secreted by the intestines. Several studies have shown that a certain dose of ghrelin is injected into the cerebral ventricle of chicks, and the results are found to strongly inhibit the feed intake of chicks [48,49]. In the HT group, a significant increase in hypothalamic ghrelin levels was observed, which is consistent with the results of a previous study [36]. There were no significant changes in other peptides except for ghrelin in the hypothalamus; no significant changes in PYY, ghrelin, and GLP-1 were observed in the intestinal tract and circulation. This is consistent with previous research [50]. It was found that heat stress significantly reduced the plasma GIP concentration of poultry and pigs [51]. The present study found that heat stress significantly reduced the GIP concentration in the jejunum of broilers, and the serum GIP concentration had a decreasing trend. This is supported by previous results. The decreased GIP concentration may be caused by the decreased number of intestinal K cells or the decreased GIP expression in K cells. Another possible explanation for this is that blood glucose was significantly reduced because GIP secretion is glucose-dependent [52].

The gut microbiota participating in host metabolism is frequently reported on [53,54,55]. According to various studies, such a process is associated with bacterial metabolites, such as SCFAs [56,57]. In the current study, the proportion of acetate-producing bacteria, such as *Ruminococcus* and *Clostridium,* increased, leading to an increase in acetate production. Studies have found that oral prebiotics may affect the secretion of related gastrointestinal hormones, such as GLP-1 and ghrelin, through the short-chain fatty acids produced by microbiota fermentation [55]. SCFAs have also been shown to cross the blood–brain barrier, directly into the brain, to control feed intake [58]. Furthermore, it was reported that SCFAs can stimulate mouse intestinal L cells to secrete PYY, CCK, and ghrelin, by activating G protein coupled receptors, such as Gpr41 [59]. A recent study suggested that SCFAs inhibit the secretion of GIP by activating the FFAR3 receptor of K cells [60]. This evidence implies that the increased CCK and decreased GIP relate to the SCFAs in the current study. Therefore, it is not surprising that the increase in CCK secretion by L cells and the decrease in GIP secretion by K cells may be due to the increased acetate content in the cecal contents.

## 5. Conclusions

This paper argues that the poor production performance of broilers under heat stress may be related to SCFAs fermented by gut microbiota involvement in regulating metabolic disorders. Our data will provide valuable insight for future studies on humans in tropical environments who suffer from metabolic disorders, such as diabetes and obesity.

## Figures and Tables

**Figure 1 animals-11-01286-f001:**
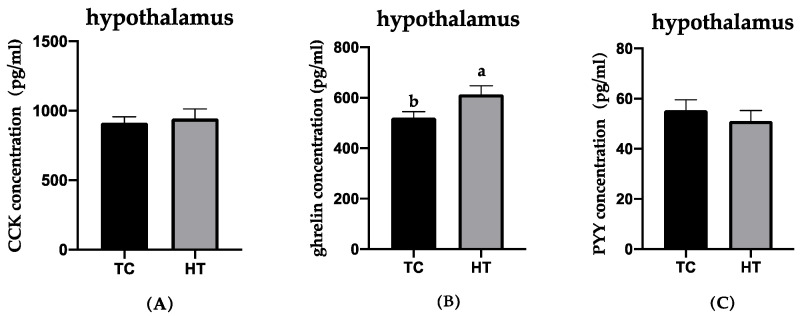
Effects of high temperatures on CCK, ghrelin, and PYY concentrations in the hypothalamus. (**A**), cholecystokinin; (**B**), ghrelin; (**C**), peptide YY. a,b, Means with different letters within columns indicates significant differences (*p* < 0.05).

**Figure 2 animals-11-01286-f002:**
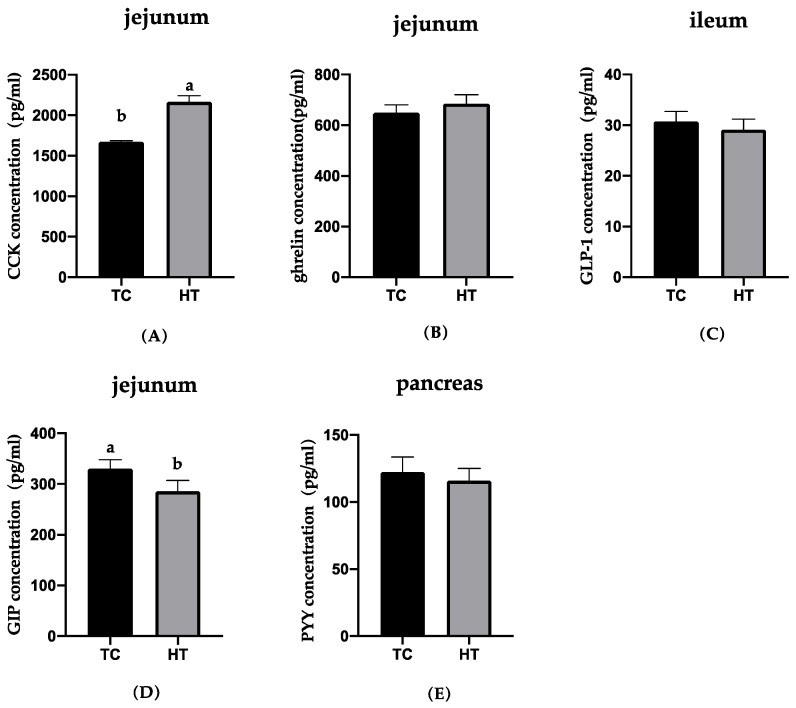
Effects of high temperature on concentrations of CCK, ghrelin, GIP in jejunum, GLP-1 in ileum and PYY in pancreas. (**A**), cholecystokinin; (**B**), ghrelin; (**C**), glucagon-like peptide-1; (**D**), glucose-dependent insulinotropic polypeptide; (**E**), peptide YY. a,b, Means with different letters within columns indicates significant differences (*p* < 0.05).

**Figure 3 animals-11-01286-f003:**
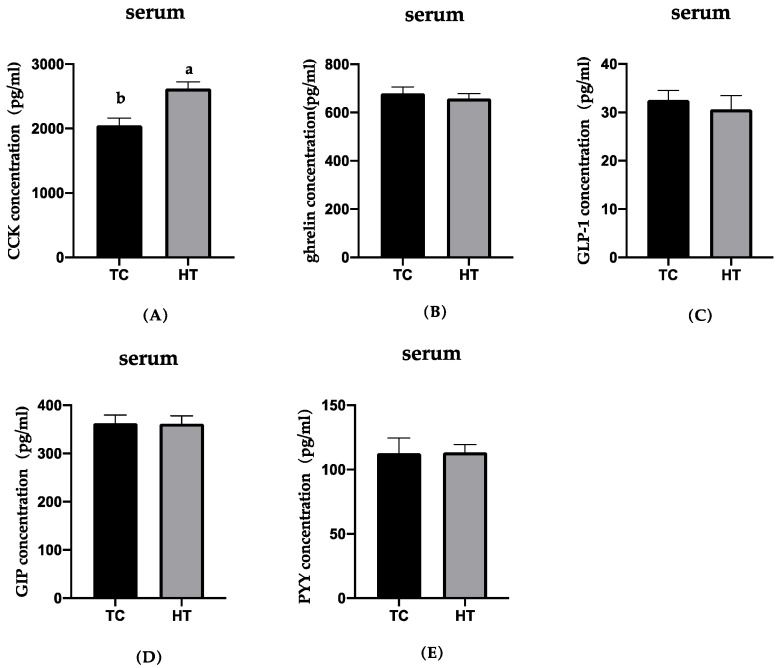
Effects of high temperature on concentration of CCK, ghrelin, GLP-1, GIP and PYY in serum. (**A**), cholecystokinin; (**B**), ghrelin; (**C**), glucagon-like peptide-1; (**D**), glucose-dependent insulinotropic polypeptide; (**E**), peptide YY. a,b Means with different letters within columns indicates significant differences (*p* < 0.05).

**Figure 4 animals-11-01286-f004:**
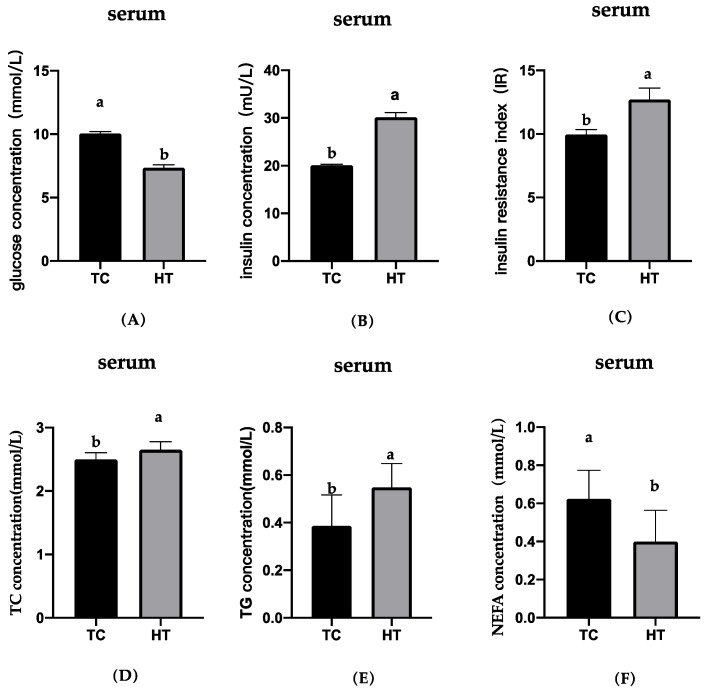
Effect of high temperature on blood glucose, insulin, insulin resistance index, TC, TG, and NEFA in serum. (**A**), blood glucose; (**B**), insulin; (**C**), insulin resistance index; (**D**), total cholesterol; (**E**), triglyceride; (**F**), non-esterified fatty acid. ^a,b^ means with different letters within columns indicates significant differences (*p* < 0.05).

**Table 1 animals-11-01286-t001:** Composition and nutrient levels of the basal diet.

Items	Content (%)
Ingredients	
Corn	56.51
Soybean meal	35.52
Soybean oil	4.50
Na Cl	0.30
Limestone	1.00
Ca HPO_4_	1.78
d L-Met	0.11
Premix (1)	0.28
Total	100.00
Nutrient levels (2)	
ME/(MJ/Kg)	12.73
CP	20.07
Ca	0.90
AP	0.40
Lys	1.00
Met	0.42
Met + Cys	0.78

(1) Premix provided the following per kg of the diet: VA 10,000 IU, VD3 3400 IU, VE 16 IU, VK3 2.0 mg, VB1 2.0 mg, VB2 6.4 mg, VB6 2.0 mg, VB12 0.012 mg, pantothenic acid calcium 10 mg, nicotinic acid 26 mg, folic acid 1 mg, biotin 0.1 mg, choline 500 mg, Zn(ZnSO_4_·7H_2_O) 40 mg, Fe(FeSO_4_·7H_2_O) 80 mg, Cu(CuSO_4_·5H_2_O) 8 mg, Mn(MnSO_4_·H_2_O) 80 mg, I(KI) 0.35 mg, Se(Na_2_SeO_3_) 0.15 mg. (2) Calculated values.

**Table 2 animals-11-01286-t002:** Effects of thermal environment on performance indices of broilers during experiment period.

	Treatments			
Item	TC	HT	SEM	*p* Value
IABW (g)	1427.31	1432.10	19.35	>0.05
FABW (g)	2840.22 ^a^	2567.78 ^b^	24.32	<0.05
ADG (g/d)	80.58 ^a^	70.70 ^b^	5.03	<0.05
ADFI (g/d)	157.51 ^a^	148.95 ^b^	7.16	<0.05
FCR (g/g)	1.95 ^b^	2.11b ^a^	0.11	<0.05
CT (°C)	41.49 ^b^	42.66 ^a^	1.28	<0.05

Values are means ± SEM. TC, thermoneutral control group; HT, high temperature group; IABW, initial average body weight (29 d); FABW, final average body weight (43 d); ADG, average daily gain; ADFI, average daily feed intake; FCR, feed conversion rate; CT, core temperature. ^a,b^ Means within the same line with different superscript differ significantly.(*p* < 0.05).

**Table 3 animals-11-01286-t003:** Effects of the thermal environment on cecum digesta microbiota composition.

Level	Species Name	Treatments		SEM	*p* Value
		TC	HT		
	*Firmicutes* (%)	89.33	87.49	2.18	>0.05
phylum	*Bacteroides* (%)	5.69	5.18	0.74	>0.05
	*Actinomycetes* (%)	3.78	6.50	1.12	>0.05
family	*Ruminococcaceae* (%)	51.32	51.14	1.25	>0.05
	*Lachnospiraceae* (%)	18.68 ^a^	10.15 ^b^	0.80	<0.05
	*Christensenellaceae* (%)	1.20 ^a^	0.51 ^b^	0.07	<0.05
	*Peptococcaceae* (%)	0.21 ^b^	0.54 ^a^	0.05	<0.05
	*Faecalibacterium* (%)	28.32	31.63	1.18	>0.05
genus	*Romboutsia* (%)	6.91	11.29	0.45	>0.05
	*Ruminococcus* (%)	0.08 ^b^	0.15 ^a^	0.02	<0.05
	*Clostridium* (%)	2.55 ^b^	5.16 ^a^	0.40	<0.05
	*Faecalibacterium* (%)	28.32	31.63	1.18	>0.05

Values are means ± SEM. TC, thermoneutral control group; HT, high temperature group. ^a,b^ Means within the same line with different superscript differ significantly (*p* < 0.05).

**Table 4 animals-11-01286-t004:** Effects of thermal environment on the content of SCFAs in the cecum.

Item	Treatments		SEM	*p* Value
	TC	HT		
acetate (μg/mL)	572.9 ^b^	741.1 ^a^	16.79	<0.05
propionic acids (μg/mL)	56.45	63.96	5.89	>0.05
butyric acids (μg/mL)	262.9	252.7	29.71	>0.05

Values are means ± SEM. SCFAs, short chain fatty acids; TC, thermoneutral control group; HT, high temperature group. ^a,b^ Means within the same line with different superscript differ significantly. (*p* < 0.05).

## Data Availability

Not Applicable.

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
