# Peer review of "Effects of Heat Stress on Gut-Microbial Metabolites, Gastrointestinal Peptides, Glycolipid Metabolism, and Performance of Broilers"

_animals, 2021, doi:10.3390/ani11051286_

Round 1
Reviewer 1 Report
The aim of this study was to evaluate the effects of heat stress on gut-microbial metabolites, gastrointestinal peptides, glycolipid metabolism and performance of broilers. The number of birds used in the experiment is sufficient, the test methods used are correct. The discussion is well carried out and exhausting. References well chosen. Before publishing in Animals, the paper requires additions and corrections. The list of proposed changes is given below:
General comments:
Please place the Tables and Figures in the Results chapter in the appropriate place related to the description. Delete subchapter 3.2. “Figures and Tables”. See other articles in Animals.
In the "References" chapter abbreviated name journal and volume must be "in italic", the year of publication must be in bold- see instructions for authors.
Use a lowercase "p" in italic, instead of a capital P to denote relevance throughout the article.
Detailed comments:
L22 “132 male one-day old” (see L81)?
L24 average daily gain (ADG), average daily feed intake (ADFI) instead of current form (because for the first time in the text)
L25 “feed conversion” decreased but FCR increased
L35 NEFA increased significantly by Figure 4F - not decreased
L43 global poultry meat production ”not global chicken production” because global chicken meat production was 118.0 million tons in 2019 not 130.57 million tons according to FAOSTAT
L54 "nonestrified fatty acid (NEFA)" instead of current form
L82, please specify dimensions of a single cage, cage per cm2
L83 please provide information about the lighting program (length, intensity, type of light)
L89 “full name (CAAS)” instead of current form
L96 + Please provide the name and type of balance, data of the balance manufacturer, accuracy of measurement used to determine BW and FI
L101 only one bird?
L103 from how many birds blood was collected from each group
L 114 20 min, space needed
L183 What was the experimental unit?
L187 FCR increased, only “feed conversion” decreased
L192 When was the end of the experiment 42 days or 49 days (28 plus 21 days ) See L87, L91, L192)
L194 Dot after (p < 0.05)
L197-200 Repetition of the description of test results contained in lines 192-197
L205 In table 2 there are only data for family and genus, not phyla
L207 In table 2 no data for phylus level: Firmicutae, Bacteroides, Actinonycetes
L209 In table 2 no data for Family “Ruminococcaceae”
L212-214 no data for some genus levels in Table 2
L219 "and butyric acids" instead of "and butyric"
L222 According to Figure 4F, NEFA is significantly higher in HT than TC group - the description is not consistent with the data
In table 1, I propose to include data for "BW 29 d", "BW 49 d"
In table 2 all data for phylus, family, genus are included in the description of the results - the Results chapter
In Q table 3, elimination of the marking "a" for non-significance for “propionate” and “butyrate”,
L250 add explanation "CT - core temperature"
L255 In the title "cecum digesta" instead of current form?
L256 add an explanation "SCFA - short-chain fatty acids"
L387 S.J.H.V, A.G ... .. - no surnames
L405 J.S.S., M.O.S. etc, no surnames
Author Response
Dear Editors and Reviewers,
Thanks very much for taking your time to review this manuscript. We really appreciate all your generous comments and suggestions! Please find my itemized responses in below and my revisions in the re-submitted files.
L22 “132 male one-day old” (see L81)?
Thanks, I added male in article. But the experiment started at 28-day old, not one-day old. See L22
L24 average daily gain (ADG), average daily feed intake (ADFI) instead of current form (because for the first time in the text)
Thanks,this is my problem, it has been corrected. See L24-25
L25 “feed conversion” decreased but FCR increased
Thanks,I wrote the wrong result in reverse. See L26
L35 NEFA increased significantly by Figure 4F - not decreased
After checking the original data, it was found that the group name was reversed when drawing. See L35 and figure 4-(F)
L43 global poultry meat production ”not global chicken production” because global chicken meat production was 118.0 million tons in 2019 not 130.57 million tons according to FAOSTAT
Thank you for your correction. I calculated this data on the FAO official website, and there may be errors, because I did not find any existing data in the literature. It has been corrected. See L44
L54 "nonestrified fatty acid (NEFA)" instead of current form
Thanks, it has been corrected. See L54
L82, please specify dimensions of a single cage, cage per cm2
Thanks, I added the dimensions of a single cage. See L83
L83 please provide information about the lighting program (length, intensity, type of light)
We have added the lighting program in article, see L93.
L89 “full name (CAAS)” instead of current form
Thanks, we have added the full name of CAAS. See L90
L96 + Please provide the name and type of balance, data of the balance manufacturer, accuracy of measurement used to determine BW and FI
Thanks for your advise, I added these contents. See L99
L101 only one bird?
We selected a total of 6 chickens from each group, one for each duplicate. See L103
L103 from how many birds blood was collected from each group
The fact is that one bird was taken from each replicates. See L106
L 114 20 min, space needed
Thanks, it has been corrected. See L118
L183 What was the experimental unit?
Thank you very much,I forgot to state the experimental unit, and it should add Replicate served as the experimental unit. See L185
L187 FCR increased, only “feed conversion” decreased
The changes of FCR have been corrected. See L192
L192 When was the end of the experiment 42 days or 49 days (28 plus 21 days ) See L87, L91, L192)
It was 42 days old at the end of the experiment.
L194 Dot after (p < 0.05)
Thanks, it have been corrected.
L197-200 Repetition of the description of test results contained in lines 192-197
Thanks,this statement has been deleted.
L205 In table 2 there are only data for family and genus, not phyla
Sorry, the phyla data should be added to the table. We have added it.
L207 In table 2 no data for phylus level: Firmicutae, Bacteroides, Actinonycetes
Sorry, the phyla data should be added to the table. We have added it.
L209 In table 2 no data for Family “Ruminococcaceae”
Thanks, Ruminococcaceae has been added.
L212-214 no data for some genus levels in Table 2
Yes, this should be shown.we have added it.
L219 "and butyric acids" instead of "and butyric"
Thanks, it has been corrected. See L244
L222 According to Figure 4F, NEFA is significantly higher in HT than TC group - the description is not consistent with the data
After checking the original data, it was found that the group name was reversed when drawing. In fact, the level of NEFA was drop significantly.
In table 1, I propose to include data for "BW 29 d", "BW 49 d"
Thanks,it has been added.
In table 2 all data for phylus, family, genus are included in the description of the results - the Results chapter
Thank you, it has been checked and corrected.
In Q table 3, elimination of the marking "a" for non-significance for “propionate” and “butyrate”,
Thank you, it has been deleted.
L250 add explanation "CT - core temperature"
Thanks,it has been added. See L198
L255 In the title "cecum digesta" instead of current form?
Thank you, it has been corrected. See L237
L256 add an explanation "SCFA - short-chain fatty acids"
Thanks,it has been added.
L387 S.J.H.V, A.G ... .. - no surnames
Thank you, it has been corrected. See L379
L405 J.S.S., M.O.S. etc, no surnames
Thank you, it has been corrected. See L398

Reviewer 2 Report
The manuscript describes a straightforward experiment with 2 groups of broilers kept either at ambient temperature (21 C) or under conditions defined as heat stress (31C) for a period of 21 days. The conidiations resulted in significant differences in feed intake, feed utilization and average weight gains and modified various parameters of intestinal function related to feed intake and overall glucose and fat metabolism.
Comments:
Line 262-264: This statement is not entirely current, as not alle mentioned parameters are significantly changes. Pleas rephrase.
Line 287 onwards: The discussion is based on the assumption that NEFA concentrations are decreased under heat stress, while in figure 4 f an increase is depicted. In table 3 an increase in acetate acid and propionate acid is reported. Butyrate is increased and these results should be discussed also in relation to the changes in the microbiota. In more detail.
Lin 306-310: both sentences are incomplete. Please correct.
Concussion section: This section needs extensive editing, as several sentences are incomplete.
Author Response
Dear Editors and Reviewers,
Thanks very much for taking your time to review this manuscript. We really appreciate all your generous comments and suggestions! Please find my itemized responses in below and my revisions in the re-submitted files.
Line 262-264: This statement is not entirely current, as not all mentioned parameters are significantly changes. Pleas rephrase.
Thank you for pointing this out, we have made adjustments. See L265-267
Line 287 onwards: The discussion is based on the assumption that NEFA concentrations are decreased under heat stress, while in figure 4 f an increase is depicted. In table 3 an increase in acetate acid and propionate acid is reported. Butyrate is increased and these results should be discussed also in relation to the changes in the microbiota. In more detail.
I’m sorry, and after checking the original data, it was found that the group name was reversed when drawing,so in fact, our result of NEFA was still decreased. The correct figure has been attached. Because I used inappropriate superscripts in the table 3, in fact, propionic acid and butyric acid did not change significantly.
Lin 306-310: both sentences are incomplete. Please correct.
We have made a revision in these sentences. See L309-311
Concussion section: This section needs extensive editing, as several sentences are incomplete.
Do you mean the conclusion section? We have revised the statements in conclusion section. See L353-356

Reviewer 3 Report
Although effects of heat stress on animals included broiler has been reported by several studies, this paper investigated insight into gut-microbial metabolites, gastrointestinal peptides, glycolipid metabolism. The manuscript is interesting and valuable in some extent and could be published. However, the authors are required to do some changes as my following comments:
- L112-114: make consistent for the term ELISA. L113: centrifuged for 20 min (2000r-300r//min) should be rpm.
- L159-160: "then centrifuge at 4 ℃ overnight at 10000rpm for 10min. Finally it was centrifuged overnight or for 10 min?
- Check spelling for mL, rpm, min (a space before these term)
- L165: what is UL?, then in L171 it becomes μ L?
- L172: do not understand what the author means "holding for 1 h min"
- Section 3.2. Figures and Tables should be removed. I think it still remain from journal template.
- All tables figures should be placed immediately after the text that they were mentioned (as the guidance from journal)
- The current titles of all figures should be removed and need to be replaced by name of tissue because they were included in vertical axis title. For example, CCK should be replaced by hypothalamus.
- In all tables, the data should be presented as the mean plus pooled SEM rather than mean plus individual SEM
- Table 1: the unit of FCR (%) seems incorrect and the parameter of CT need to be explained.
- Table 2:the units of species name need to be included
- All figures need to increase the size
Author Response
Dear Editors and Reviewers,
Thanks very much for taking your time to review this manuscript. We really appreciate all your generous comments and suggestions! Please find my itemized responses in below and my revisions in the re-submitted files.
L112-114: make consistent for the term ELISA. L113: centrifuged for 20 min (2000r-300r//min) should be rpm.
Thank you, it has been corrected. See L116-118
L159-160: "then centrifuge at 4 ℃ overnight at 10000rpm for 10min. Finally it was centrifuged overnight or for 10 min?
Sorry, it should be“overnight at 4 ℃, then centrifuge at 10000rpm for 10min”. See L164
Check spelling for mL, rpm, min (a space before these term)
It has been corrected.
L165: what is UL?, then in L171 it becomes μ L?
It should be μL,I made a mistake. See L169
L172: do not understand what the author means "holding for 1 h min"
Sorry, I made a mistake writing, it has been corrected. See L176
Section 3.2. Figures and Tables should be removed. I think it still remain from journal template.All tables figures should be placed immediately after the text that they were mentioned (as the guidance from journal)
Thanks,The charts has been moved to the correct position.
The current titles of all figures should be removed and need to be replaced by name of tissue because they were included in vertical axis title. For example, CCK should be replaced by hypothalamus.
Thanks,and the titles of figures have replaced by name of tissue. We have inserted the corrected figures in the article.
In all tables, the data should be presented as the mean plus pooled SEM rather than mean plus individual SEM
Thanks,and we have replaced current form with mean plus pooled SEM.
Table 1: the unit of FCR (%) seems incorrect and the parameter of CT need to be explained.
Thank you for your rectify and we have corrected.
Table 2:the units of species name need to be included
Thanks,we have added the units of species name.
All figures need to increase the size
We have enlarged all figures.

Round 2
Reviewer 1 Report
The aim of this study was to evaluate the effects of heat stress on gut-microbial metabolites, gastrointestinal peptides, glycolipid metabolism and performance of broilers. The number of birds used in the experiment is sufficient, the test methods used are correct. The discussion is well carried out and exhausting. References well chosen. Before publishing in Animals, the paper requires additions and corrections. The list of proposed changes is given below:
General comments:
From the text on pages 3 to 11, please use the “justify” function
In references: Short name of the journal and volume must be in italics
The year of publication must be in bold
See instructions for authors:
Author 1, A.B .; Author 2, C.D. Title of the article. Short name of the journal Year, volume, page range.
Throughout the article (p < 0.05), a space after "p" and after "<"
Detailed comments:
L82 space after "three-layer"
L89 "full name (CAAS)" in place of its current form
L96 + information is needed on nutrition program, the form of the mixture (granules, crumble, fine) for broiler chickens during the rearing period, CP and MJ ME levels - they have an impact on feed intake during thermal stress and BW gain,
L96 + add information about method of water administration (ad libitum, unlimited), water temperature 10-12⁰ C?, (Function: body cooling)
In table 1, a space after "IABW", "FABW"
L198 space after 29 and 43
Serum L204 GIP levels also decreased significantly? see the Figure 3D
L206 "greater than" instead of “greaterthan”
L207 "jejunum only" – “jejunum” instead of “intestine”
In table 2, a space before "(%)"
In table 3, space after "acetate"
L239-240, L247-249, please check the line spacing.
L411 “Int. J. Agric. Biol.” - there must be “dots”
L452 must be - short name journal, year of publication, volume, page range instead of the current form
Author Response
Thank you for your letter and for the reviewers’ comments concerning our manuscript entitled “Effects of heat stress on gut-microbial metabolites, gastrointes-tinal peptides, glycolipid metabolism and performance of broilers” (ID: animals-1177454). Those comments are all valuable and very helpful for revising and improving our paper, as well as the important guiding significance to our researches. We have studied comments carefully and have made correction which we hope meet with approval. Revised portion are marked. The main corrections in the paper and the responds to the reviewer’s comments are as flowing:
General comments:
From the text on pages 3 to 11, please use the “justify” function
Thanks, it has been corrected.
In references: Short name of the journal and volume must be in italics
The year of publication must be in bold
See instructions for authors:
Author 1, A.B .; Author 2, C.D. Title of the article. Short name of the journal Year, volume, page range.
Thanks,all references have been corrected.
Throughout the article (p < 0.05), a space after "p" and after "<"
Thanks,all spaces have been added.
L82 space after "three-layer"
Thanks,it has been added. See L 89
L89 "full name (CAAS)" in place of its current form
We have added (CAAS)in the paper. See L 97
L96 + information is needed on nutrition program, the form of the mixture (granules, crumble, fine) for broiler chickens during the rearing period, CP and MJ ME levels - they have an impact on feed intake during thermal stress and BW gain,
We have added information on nutrition program (Table 1). See L 91
L96 + add information about method of water administration (ad libitum, unlimited), water temperature 10-12⁰ C?, (Function: body cooling)
We have added the information about method of water administration(ad libitum)and water temperature is equal to room temperature. See L 90
In table 1, a space after "IABW", "FABW"
Thanks,they have been added. See Table 2.
L198 space after 29 and 43
Thanks,they have been added. See L 328
Serum L204 GIP levels also decreased significantly? see the Figure 3D
Sorry,we made a mistake,in fact there is not a significant change of GIP in serum. We have corrected it. See L334
L206 "greater than" instead of “greaterthan”
Thanks, it has been corrected. See L336
L207 "jejunum only" – “jejunum” instead of “intestine”
Thanks,we have added in ileum(GLP-1) and in pancreas(PYY). See L337
In table 2, a space before "(%)"
Thanks,they have been added. See Table3
In table 3, space after "acetate"
Thanks,it has been added. See Table4
L239-240, L247-249, please check the line spacing.
Thanks, it has been corrected. See L382-383 L394-396
L411 “Int. J. Agric. Biol.” - there must be “dots”
Thanks, dots have been added. See L767
L452 must be - short name journal, year of publication, volume, page range instead of the current form
Thanks, it has been corrected. See L817
